# Addressing Parental Vaccine Hesitancy towards Childhood Vaccines in the United States: A Systematic Literature Review of Communication Interventions and Strategies

**DOI:** 10.3390/vaccines8040590

**Published:** 2020-10-08

**Authors:** Olivia Olson, Corinne Berry, Nirbhay Kumar

**Affiliations:** 1Department of Global Health, Milken Institute School of Public Health, The George Washington University, Washington, DC 20052, USA; 2CommunicateHealth, Rockville, MD 20850, USA; corinne@communicatehealth.com

**Keywords:** vaccine hesitancy, childhood vaccines, health communication, messaging, vaccine literacy, interventions, strategies, campaigns

## Abstract

Parental vaccine hesitancy is becoming an increasingly important public health concern in the United States. In March 2020, an assessment of the latest CDC National Immunization Survey data found that more than one-third of U.S. children between the ages of 19 and 35 months were not following the recommended early childhood immunization schedule. Furthermore, a 2019 national survey found that approximately 1 in 4 parents reported serious concerns towards vaccinating their children. Vaccine hesitancy is now associated with a decrease in vaccine coverage and an increase in vaccine-preventable disease outbreaks and epidemics in the United States. Many studies have focused on understanding and defining the new socio-medical term, vaccine hesitancy; few have attempted to summarize past and current health communication interventions and strategies that have been successful or unsuccessful in tackling this growing phenomenon. This systematic literature review will attempt to aid public health professionals with a catalogue of health communication interventions and strategies to ultimately address and prevent parental vaccine hesitancy in the long term. Out of 1239 search results, a total of 75 articles were included for analysis, ranging from systematic reviews, quantitative surveys, and experimental designs to ethnographic and qualitative studies. For the presentation of results, a taxonomy was used to organize communication interventions according to their intended purpose. The catalogue of interventions was further broken down into specific components and themes that were identified in the literature as essential to either the success or failure in preventing and addressing parental vaccine hesitancy towards childhood vaccines.

## 1. Introduction

### 1.1. Overview of Parental Vaccine Hesitancy Towards Childhood Vaccines in the U.S.

Vaccination is widely considered one of the greatest achievements in public health [1,2]. Vaccination programs have contributed significantly to the decline in mortality and morbidity from various infectious diseases. Most notably, vaccination has been credited with the elimination of polio in the Americas and eradication of smallpox worldwide [3]. Despite being recognized as one of the most successful public health measures for controlling and preventing disease, there has been an upward trend in the perception that vaccination is unsafe and unnecessary by a growing number of individuals in the United States (U.S.) [4]. As a result, hesitancy towards vaccination is beginning to threaten historical achievements that have been made in the U.S. to eliminate and reduce the burden of many infectious diseases that have afflicted humanity for centuries.

One particular concern in the U.S. is parental vaccine hesitancy towards childhood vaccines. Parental vaccine hesitancy has serious consequences for children and surrounding communities. A 2019 national survey found that approximately 1 in 4 parents reported serious concerns towards vaccinating their children [5]. In 2018, the Centers for Disease Control and Prevention (CDC) reported that while overall childhood immunization rates remained high during the time period 2012 to 2017, there continued to be an increase in the number of children with no vaccines at age 24 months [6]. More recently, in March 2020, an assessment of the latest CDC National Immunization Survey data found that more than one-third of U.S. children between the ages of 19 and 35 months were not following the recommended early childhood immunization schedule [7]. Concurrent with the increase in parental vaccine hesitancy is an increase in parents choosing to opt out their children from vaccinations required for school entry by obtaining non-medical vaccine exemptions [8]. The consequences of the increase in parental vaccine hesitancy over the last several years can most notably be seen in the recent U.S. measles outbreaks. Measles is one of the most contagious infectious diseases. Measles was declared eliminated from the U.S. in 2000 due to the nation’s effective vaccination program and strong public health system for detecting and responding to infectious disease outbreaks. Today, outbreaks of measles are occurring in the U.S. at an alarming rate, as unvaccinated groups are exposed to occasional outbreaks of the measles virus from outside the country [9]. A notable epidemic was the December 2014 measles outbreak among 125 children in Disneyland [10]. Outbreaks of measles have continued in the U.S. over the last two decades. The first spike in measles cases was in 2008 with 64 confirmed reported measles cases, followed by 667 reported cases in 2014, 375 reported cases in 2018, and a record-breaking outbreak of 1,282 cases in 2019 [11]. Large measles outbreaks among children are now occurring alongside an increasing number of parents either delaying or refusing to vaccinate their children against measles [12].

While opposition to vaccination has existed as long as vaccination itself, the subject of vaccines is especially contentious today [13]. There are numerous messages circulating widely both in support of and against childhood vaccines in the U.S. [14,15]. Parents seeking information about childhood vaccines have ample opportunity for exposure to false and misleading information about vaccines through traditional media outlets and interactive Web 2.0 and social media outlets online [16]. The plethora of immunization misinformation online buries science-based information and greatly affects health care professionals’ communication abilities and public immunization programs [16,17]. Misinformation and controversies about childhood vaccines tend to focus on questioning the safety and efficacy of specific vaccines [18,19]. One of the more prominent controversies has been the claim that the measles, mumps and rubella (MMR) vaccine causes autism. This claim originated from the 1998 Lancet article published by Andrew Wakefield, which was later retracted. While there is no scientific evidence that the MMR vaccine causes autism, the 1998 article had already caused fear, confusion and distrust over the safety of the vaccine that continues to this day [20]. Data from the last decade show that vaccine hesitancy among parents is continuing to increase as some parents begin to follow alternative childhood vaccination schedules, some become selective about which vaccines to give their child, and others refuse vaccines all together [19,21,22,23,24,25,26,27].

The decrease in childhood vaccine coverage and increase in childhood vaccine-preventable disease outbreaks and epidemics in the U.S. and around the world has been associated with what is now known as “vaccine hesitancy” [28]. In 2019, vaccine hesitancy was named one of the top 10 public health threats by the World Health Organization [28]. According to the WHO’s Strategic Advisory Group of Experts on Immunization (SAGE), vaccine hesitancy refers to a delay in the acceptance or refusal of vaccines despite the availability of vaccination services. Vaccine hesitancy is complex and context specific, varying across time, place, and vaccines. It is influenced by factors such as complacency, convenience, and confidence [29]. Vaccine hesitancy has been an emerging term in the socio-medical literature and describes vaccine decision-making or a lack of “vaccine confidence” and results in either delay or complete refusal [30,31]. Reasons for vaccine hesitancy are complex, and there are far more parents who are hesitant towards vaccines than outright refusing all vaccines [32]. A large number of conceptual models have been developed for grouping vaccine hesitancy determinants, which have brought both clarity and confusion to understanding the determinants of this phenomenon [31].

With the growing rates of vaccine hesitancy in the U.S., public health authorities are looking for effective communication interventions and strategies to promote vaccination, combat anti-vaccination messaging, and address vaccine hesitancy. To date, little is known about key health communication interventions and strategies that could be successful in addressing and preventing childhood vaccine hesitancy in the long term. What is known is that thus far, the majority of interventions to combat vaccine hesitancy have been educational and focused on a “knowledge-deficit” approach, assuming that vaccine-hesitant individuals will change their mind if given proper information. [24,33,34]. However, research has shown that vaccine hesitancy is more complex, involving emotional, cognitive, cultural, spiritual, social, and political factors [24,29,30,35]. Research has also shown that it matters which message techniques are used, and which messengers deliver the vaccine information [19,36]. Furthermore, the spread of false and misleading information about vaccines has led many parents to question the efficacy and safety of vaccines, dramatically impacting public understanding and trust in vaccines [24,25,37,38].

### 1.2. Vaccines, Vaccination, and Immunization

Vaccines help the body’s immune system fight off infections caused by bacterial, parasitic, and viral pathogens capable of causing serious, debilitating, and often deadly diseases. Since the introduction of the first vaccine, vaccination has been able to control many major diseases: smallpox, measles, mumps and rubella, diphtheria, pertussis, tetanus, yellow fever, and poliomyelitis [39,40,41,42,43]. Vaccination leads to the development of long-lasting and specific immunity against the infectious pathogen without any serious adverse events. Vaccination can be divided into two categories, active and passive vaccination approaches. In passive vaccination, preformed antibodies are administered just before or around the time of exposure. On the other hand, in the active vaccination approach, vaccines stimulate the immune system to produce either specific humoral (antibodies) and cellular immune responses, or both. Vaccines can be further divided into different categories: live attenuated vaccines, killed vaccines, subunit vaccines, nucleic acid (DNA and mRNA) vaccines, and vectored vaccines [39,40]. It is important to emphasize that an infection by a pathogen also leads to activation of the body’s immune system to elicit specific immune responses capable of providing partial or complete immunity to fight off the ongoing infection or future infections [39,40,44]. The significance of specific immune responses also lies in the fact that active vaccination leads to the generation of immunological memory, which helps the body to respond quickly and in a robust manner to prevent disease and death. Thus, a vaccine activates the body’s immune system upon administration (the process of vaccination), resulting in the generation of effective immunity (the process of getting immunized) to an infection or the disease caused by a pathogen [39,40,44].

### 1.3. Components of Vaccines

Individual components included in a vaccine formulation either participate during the activation of appropriate innate and adaptive immune responses to provide immunity or serve to stabilize the vaccine and maintain safety for the long lasting effectiveness of the vaccines (Table 1) [45,46]. The components that help the immune system respond and build immunity to a specific disease include antigens and adjuvants. Antigens can be small amounts of weakened, attenuated or killed pathogen that can cause disease or a defined macromolecular component (protein, carbohydrate, or nucleic acid). Adjuvants are immunomodulatory components that activate the innate immune system and help the immune system respond strongly to a vaccine to mount an appropriate and desired immune response [45]. The other components that keep vaccines safe and stable include preservatives and stabilizers. Preservatives protect the vaccine from external contaminations, and stabilizers such as sugar or gelatin enhance the stability of vaccines during different phases of production, storage, and shipping.

The components of newly licensed and existing vaccines are continuously reviewed and undergo an exhaustive safety and toxicity evaluation [46]. It is only upon such regulatory approvals by the FDA that new or modified existing vaccines move forward to licensure and recommendation for use in the U.S.A. by the CDC. Thus, monitoring vaccine safety and effectiveness remains an ongoing process for any vaccine intended for public use. The CDC routinely publishes the written recommendations for vaccinating U.S. children and sets the immunization schedules based on recommendations from the Advisory Committee on Immunization Practices (ACIP) and approval by the American Academy of Pediatrics (AAP), the American Academy of Family Physicians (AAFP), and the American College of Obstetricians and Gynecologists [47]. Table 2 outlines the current CDC-recommended vaccination schedule for children. Additional information about specific types of vaccines and vaccine-preventable diseases can be found on the CDC website [43,46,48].

The goal of this systematic literature review is to aid public health authorities in developing effective strategies to address parental vaccine hesitancy towards childhood vaccines in the U.S. The definition, scope, and determinants of vaccine hesitancy differ greatly for different age groups, children, adolescents, young adults, and older adults. They also differ depending on the vaccine such as the MMR vaccine, the human papilloma virus (HPV) vaccine, the polio vaccine, the pertussis vaccine, and the seasonal flu vaccine, to name a few. The scope and determinants of vaccine hesitancy differ greatly for high-income countries (HICs) and lower- and middle-income countries (LMICs). This review will therefore focus specifically on parental vaccine hesitancy towards childhood vaccines in the U.S. The majority of discussion will focus on one main objective: exploring key health communication interventions and strategies that could be successful in addressing and preventing parental vaccine hesitancy towards childhood vaccines in the long term. There will also be discussion on the definition of vaccine hesitancy and the key determinants of vaccine hesitancy for childhood immunizations.

## 2. Materials and Methods

A systematic literature review was conducted following a structured search and screening process inspired by the Preferred Reporting Items for Systematic Reviews and Meta-Analysis (PRISMA) guidelines (Figure 1). The search was performed in PubMed, PsychINFO, ProQuest, Cochrane Library, and Communication and Mass Media Complete databases, targeting publications from January 2008 to October 2019. Additional articles were identified through Google Scholar and other sources. The search strings included a variety of combinations of terms related to “vaccine hesitancy”, “communication”, and “intervention”. The full list of search terms is included in Table 3. The results were filtered to include articles that were full-text, available in English, and predominately from the U.S. International publications from a few high-income, English-dominant countries including the U.K., Australia, and Canada were also included. The databases were initially searched for articles related to the main objective: exploring key communication interventions and strategies that can be successful in addressing and preventing parental vaccine hesitancy towards childhood vaccines in the long term. The inclusion criteria required the focus of the study to be on parental vaccine hesitancy around childhood vaccines. Articles with a primary focus outside of parental vaccine hesitancy and childhood vaccines were excluded. Articles that focused on annual/seasonal vaccines (i.e., influenza), adolescent immunizations (i.e., HPV or meningitis), or adult and older-adult vaccines (i.e., shingles) were also excluded. Articles were also excluded if their focus was on strategies and interventions to solely reduce immunization pain or if the intervention broadly focused on improving immunization rates and not addressing vaccine hesitancy. This systematic literature review followed a four-step screening process. First, reference manager software was used to combine and review the search results and eliminate any duplicates. Studies with multiple publications were narrowed down to the most applicable or informative article. Second, the articles were screened by two reviewers for relevance by title and abstract. The third screening process ensured that articles were available in full text and met the inclusion and exclusion criteria. Lastly, articles were excluded if not relevant or adding new information to the main objective. Relevant data were collected in Excel, including the study design and methods, sample size, target audience, country of publication, study results, and study implications. A total of 75 articles were included, ranging from systematic reviews, quantitative surveys, and experimental designs to ethnographic and qualitative studies.

## 3. Results

### 3.1. Defining Vaccine Hesitancy

Several definitions have been proposed for the emerging term “vaccine hesitancy”. The WHO SAGE Working Group’s definition of vaccine hesitancy is the most commonly cited in the literature [4,23,24]. MacDonald, in 2015, highlighted the importance of one practical definition of vaccine hesitancy for immunization program managers, policy makers, clinicians, and researchers to effectively understand and address issues around vaccine hesitancy [4,31]. According to the WHO SAGE Working Group on Vaccine Hesitancy, it refers to a delay in the acceptance or refusal of vaccines despite the availability of vaccine services. Vaccine hesitancy is complex and context specific, varying across time, place, and vaccines. It includes factors such as complacency, convenience, and confidence [29]. The WHO SAGE Working Group, along with several other social scientists, further define vaccine hesitancy on the continuum of those who fully accept with few doubts, those who delay the vaccination schedule, and those who completely reject vaccines (Figure 2) [29,31,49]. Several tools have been developed to define and measure vaccine hesitancy including the Vaccine Confidence Scale, the Global Vaccine Confidence Index, and the Vaccine Hesitancy Scale [29,50,51,52,53]. The most widely used tool specifically designed to measure parental vaccine hesitancy is the Parent Attitudes about Childhood Vaccines (PACV) survey, which was developed in 2011 [54,55]. The PACV survey is a short and self-administered validated survey that measures attitudes and beliefs about immunization prior to health care appointments to allow health care providers (HCP) to be informed about any hesitancy parents may have towards vaccines and to assist HCP in adapting their messages and communication strategies [56,57,58,59].

### 3.2. Key Determinants of Childhood Vaccine Hesitancy

The complexity of vaccine hesitancy and its determinants has led to a number of conceptual models for grouping vaccine hesitancy determinants [24,31,61,62,63]. The more well-known and widely referenced model “Vaccine Hesitancy Determinants Matrix” outlines factors influencing the decision-making behavior to accept, delay, or reject some or all vaccines under three categories: contextual, individual and group, and vaccine/vaccination-specific influences (Table 4) [29].

### 3.3. Key Communication Interventions and Strategies That Address Vaccine Hesitancy

A select number of communication interventions found from the literature are categorized below as per Kaufman’s comprehensive taxonomy for organizing interventions for childhood vaccination in routine and campaign contexts [64]. The interventions were categorized as those that inform and educate, remind or recall, enhance community ownership, teach skills, provide support, facilitate decision making, and enable communication.

#### 3.3.1. Inform and Educate

Gowda et al. [65] evaluated individually tailored versus untailored education materials for improving vaccination intention and positive vaccination attitudes for the MMR vaccine among parents who were surveyed and identified on a scale of vaccine hesitancy. The researchers tested web-based, individually tailored education material at four levels: image tailoring to parents’ race, content tailoring to specific vaccine-related concerns, experiential tailoring for parents’ past experiences, and name tailoring to the child’s name for messaging content. The tailored materials had a significantly greater impact on positive vaccination intentions and attitudes towards vaccination compared to untailored information provided through general educational web pages [65]. In another study, Jolley et al. [66] studied the effectiveness of attempting to counter and correct conspiracy arguments about childhood vaccines. It was found that anti-conspiracy arguments were not effective in increasing a person’s intention to vaccinate unless an individual was exposed to an anti-conspiracy theory message before providing the original conspiracy theory message.

#### 3.3.2. Remind or Recall

Frew et al. [67] conducted a systematic literature review of 34 studies that used parental reminder and recall interventions based on the use of postcards, letters, the telephone, or a combination to improve early childhood immunization uptake. Researchers found that the majority of reminder and recall systems (79%) increased the likelihood of immunization uptake from study baselines by varying degrees, anywhere from 9% to 55%. In the same review, it was found that the use of tailored calendars that aim to encourage the immunization of children up to 24 months resulted in an increase in the immunization rate by 66% in the intervention group compared to 47% among the controls. In this study, calendars were tailored to the age of the baby, and included tailored pictures and messages from the family [67]. Frew et al. also discovered that several studies employed reminder and recall systems not only for vaccine-hesitant parents but for health care providers as well. It was found that utilizing a hospital’s electronic health record (EHR) to remind HCPs to discuss childhood vaccines with parents led to an improvement in early childhood immunization uptake, reflected by a reduction in missed opportunities by 36% over a 2 year period [67].

#### 3.3.3. Enhance Community Ownership

Schoeppe et al. [68] introduced an education outreach and social marketing intervention called “The Immunity Community”, which aimed to reduce vaccine hesitancy. The goals of the “Immunity Community” intervention were to 1) address parental vaccine hesitancy by empowering parents to be immunization advocates in their community, 2) improve awareness of immunization as a social norm among parents at participating sites, and 3) change those parents’ attitudes and behaviors. Surveys revealed statistically significant improvements in vaccine-related attitudes: the percentage of parents who were concerned about other parents not vaccinating their children increased from 81.2% to 88.6%, and the percentage reporting themselves as “vaccine-hesitant” decreased from 22.6% to 14.0% [68]. This study demonstrates the promise of using parent advocates as part of a community-based approach to reduce vaccine hesitancy. Attwell et al. [69] evaluated the “I Immunize” campaign, which used a similar approach to “The Immunity Community” but focused more specifically on an identity and values-based approach to change vaccination attitudes. The campaign was carried out in a community that was known for their “alternative lifestyles” and lower-than-national vaccine coverage rates and utilized local community members to be advocates for the campaign. Community members were featured on posters, billboards, local newspapers, and social media posts, to deliver co-promoted behavioral messaging such as “My name is Andrew, I use cloth nappies, I eat at wholefoods, and I immunize”. The results from a community-wide survey of 304 respondents showed that 60% of the intended audience members were positively influenced by the campaign messages, 17% were negatively impacted, and 23% declared no impact. The researchers of the campaign attributed the success of the campaign to the involvement of local community members at every stage of the intervention in order to fit the needs of their audience [69].

#### 3.3.4. Teach Skills

Henricksen et al. [58] studied the impact of physician communication training on parental vaccine hesitancy. In this two-arm cluster randomized trial carried out from March 2012 to December 2013 in four hospitals, mothers and providers were recruited from the postpartum units of the hospitals. The first component of the intervention involved a 45-min training that physicians went through, administered by a pediatrician immunization expert and health educator. The training was based on the Ask, Acknowledge, Advise communication framework. The second component was the delivery of paper materials with the branding “Let’s Talk Vaccines” detailing information on how to reinforce training messages and reach non-attenders. The third component was the delivery of 6-monthly email newsletters to physicians on a webinar version of the original training. Although this training was novel, it only had a marginal effect to no effect, and further improvements will be needed for the effectiveness of this approach [58].

#### 3.3.5. Provide Support

Daley et al. [70] introduced an internet-based platform with vaccine information and interactive social media components to improve parents’ vaccine-related attitudes. In the intervention, study participants were randomized into the following groups: a study website with vaccine information and social media components, a website with vaccine information only, or the usual care. The parents who received the vaccine information with social media components were more likely to vaccinate their child as compared with parents who received the usual care only. Furthermore, the study highlighted the importance of social media for vaccination intention [70]. Another study conducted by Gagneur et al. [71] evaluated the impact of motivational interviewing techniques to support new mothers and increase vaccination coverage for new infants. A single session delivered one time in the maternity wards led to a significant increase of 15% in the vaccination intention of mothers compared to mothers who did not receive the intervention [71].

#### 3.3.6. Facilitate Decision-Making

Williams et al. [59] conducted a randomized trial to increase acceptance of childhood vaccines by vaccine-hesitant parents. The researchers sought to explore the importance of facilitating decision-making for parents by delivering information on how to find accurate information on the Internet regarding childhood vaccines. The educational intervention for the randomized group included three components: (1) a short video developed by experts in vaccine safety and behavioral health, (2) an educational handout on common vaccine concerns, and (3) a handout with written instructions on how to find accurate medical information on the Internet. The researchers found that this brief educational intervention resulted in a short-term (~ 2-month long) benefit in improving parental attitudes towards vaccines [59].

#### 3.3.7. Enable Communication

Davis et al. [72] introduced the Immunization Education Package (IEP) intervention to improve physician/parent communication around vaccine risks/benefits during well-baby visits in which immunizations were scheduled. The IEP intervention design consisted of a vaccine contraindication screening test, a large colorful poster entitled “7 questions parents need to ask about baby shots” that was placed in examination rooms, an information sheet with answers to the seven poster questions, and an office-based in-service conducted by a local opinion leader to provide advice. Overall, IEP showed significant improvements in both written and verbal vaccine communication, and the distribution of the federally mandated Vaccine Information Statements (VIS) increased from 33% to 91% at immunization visits [72]. Berry et al. [73] introduced the Sharing Knowledge about Immunization Project (SKAI) to improve communication between patient and provider and to implement better support systems for health care providers to address vaccine-hesitant parents [73]. This project explored the “triage and treat” approach to the clinical assessment and care of vaccine hesitancy, exploring the use of a referral pathway to a specialist immunization clinic. The SKAI approach was well received and showed an increase in parents’ intents to vaccinate their children [73]. Salmon et al. [74] introduced “MomsTalkShots”, a phone, tablet, and computer app that aims to increase the uptake of maternal infant vaccines and encourage communication between mothers and pediatricians. The mobile app initially screens patients based on survey questions and offers tailored vaccination information based on the parent’s vaccine attitudes, beliefs, intentions, demographics, and source credibility. A full evaluation of this intervention is still underway, but initial results have revealed that women are finding the app helpful (95%), trustworthy (94%), interesting (97%), and clear to understand (99%) [74].

### 3.4. Key Themes Identified in the Literature for Targeting Interventions

#### 3.4.1. Audiences Targeted

The primary target group for interventions and communication strategies to reduce parental vaccine hesitancy towards childhood vaccines has been parents with children aged 0–5 years who had serious questions/concerns about the safety of vaccines but were not adamantly against vaccinating their child, i.e., parents with strong beliefs and attitudes against vaccines [26,66,75]. Another group included health care providers, family members, teachers, and community members/neighborhood coalitions who were core influencers of vaccine-hesitant parents [59,67,76]. These efforts have highlighted the importance of including first-time mothers/parents for future studies [26,71,75].

#### 3.4.2. Messenger of Vaccination Information

For interventions and communication strategies to be successful in preventing and addressing vaccine hesitancy, the “messenger” can be just as important as the “vaccine message” itself. Studies have revealed that parents are receptive to the message only if they find the messenger trustworthy and credible. It has also been suggested that the racial and/or ethnic background of the messenger (health care worker and physician) may also promote effective communication to improve vaccination rates in underrepresented minorities. It is also a common finding that health care professionals (HCP) are the preferred channels for vaccine information for parents (Table 5).

#### 3.4.3. Timing of Vaccination Information

Pregnancy represents an optimal point in time for delivering vaccination information to parents. This is the time when a parent’s intent to immunize their new infant can be most influenced and when negative attitudes about vaccination can be reshaped. This is also the time when parents are forming the foundation for all other future decisions regarding the overall health of the child [25,26,56,57,67,71,75,77,79]. Many studies suggest that education and interventions should be implemented very early in the pregnancy period since vaccine attitudes are likely formed before making vaccine decisions as parents [56,75,96]. Other studies have shown that intervening as late as postpartum can be effective [71]. Traditionally, vaccine information is provided to parents at well-child visits [57]. However, studies have shown that parents want vaccine information before and during the prenatal period and before each of the following well-child appointments [67,95]. The stepwise delivery of vaccination information, or the delivery of vaccination information at different points in time, has also been shown to increase the likelihood of childhood vaccination. This is because more frequent reminders can positively influence parents’ intent to immunize and reinforce positive beliefs and attitudes about childhood vaccination [67,79,83,97]. First-time mothers/parents have been cited as an important target population for the delivery of interventions to prevent the formation of vaccine-hesitancy attitudes and behaviors. This is an area for further research and evaluation for effectiveness [26,71,75].

#### 3.4.4. Amount of Vaccination Information

The amount of vaccination information that is currently provided is inadequate and not enough to assist parents with a decision about whether to vaccinate their child or not. Parents want longer-than-usual appointments with their HCP to discuss vaccine information. Parents want to have enough time to receive clear answers to their questions and concerns, and to be engaged in an open dialogue-based discussion about childhood vaccines. Furthermore, parents want vaccine information that is tailored to their needs and tailored to their child [25,77,84,98,99]. It is important to also consider information overload that can confuse parents when designing interventions, especially interventions with multiple components to address vaccine hesitancy. If the intervention is being delivered by well-known trusted messengers of health information such as HCP, it is also important to consider the burden of time and additional training for HCP [100].

#### 3.4.5. Content of Vaccination Information

The content of vaccination information must be tailored to the context and the target group (Table 6) [38,63,67,79,85]. For example, the needs for vaccination information are very dependent on whether parents are slightly vaccine hesitant, moderately vaccine hesitant, or severely vaccine hesitant [31,37,85]. For parents that are lower on the vaccine-hesitancy spectrum (cautious acceptors or hesitant), the content of vaccination information that is requested is more general; parents want information about vaccine safety including balanced information about the benefits and risks of vaccination. These parents also want information about how to mitigate pain related to receiving vaccines/shots [23,25,26,77]. For parents who are higher on the vaccine-hesitancy spectrum (outright decliners or late/selective with vaccines), the vaccination information that is most often requested is more specific, such as the reasons behind combined versus single vaccines, alternatives to vaccines, the science behind how vaccines are made, technical information about vaccine production and delivery, and the reasons behind vaccine policies and recommendations. In addition, several studies have suggested that the content of vaccination information for vaccine-hesitant parents must also focus on providing information about the diseases that vaccines prevent since most of today’s parents do not have first-hand experience with the more deadly diseases that are now preventable [23,25,26,77].

#### 3.4.6. Approaches to Vaccine Messaging

The effectiveness of vaccine communication strategies depends upon the message-framing techniques used [37]. Messaging in the form of storytelling with the use of gists, emotive anecdotes, and imagery has been shown to be among the most persuasive messaging strategies. Fear-based messaging, while effective in anti-vaccine messaging and the anti-vaccination movement, has been shown to be counterproductive in promoting vaccination (Table 7).

#### 3.4.7. Vaccine Misinformation and Disinformation

Successful health communication interventions and strategies must address the issues of vaccine misinformation and disinformation (Table 8). There are two basic types of false information: misinformation or inadvertently drawing conclusions based on incorrect or incomplete facts, and disinformation, the deliberate spread of falsehoods to promote an agenda. Combating misinformation and disinformation in the fight against growing vaccine hesitancy has proved to be very difficult. It is argued that misinformation and disinformation can be corrected with factual information [17]. However, several studies have shown that attempting to correct misinformation and disinformation is difficult to do and can even be counter effective and reinforce strongly held false beliefs about childhood vaccines [18,107,111]

#### 3.4.8. Vaccine Literacy

Vaccine literacy is defined as not only knowledge about vaccines but the ability to use critical and evaluation skills to seek out the right information, especially with the ever-increasing information available in the media, particularly on the Internet [35,113,114]. Parents generally find it difficult to know which vaccine information sources to trust, and are confused about where to find information that they believe is unbiased and balanced. Vaccine literacy is an important factor in reducing the negative effects of exposure to misleading reports on vaccination [38,114].

#### 3.4.9. Face-to-Face Communication Strategies

One-on-one dialogue-based communication strategies are seen as one of the most effective communication strategies for motivating parents to discuss childhood vaccines and to increase positive attitudes, behaviors, and intentions to vaccinate [25,27,34,64,73,95,97]. Parents often comment that they want to be engaged in discussion about childhood vaccinations with their HCP. They do not want to simply receive written material from their HCP. There have been a few interventions testing the effectiveness of posters in the physician’s office encouraging childhood vaccination communication between parents and HCP. These interventions have been shown to prompt more discussion between parents and HCP [27,79,115]. Social mobilization or engaging an entire community in the topic of childhood vaccination has also been shown to be an effective strategy. These are interventions that involve members of an institution, community networks, civic and religious groups, and others in a coordinated way to reach specific groups of people for dialogue and planned messaging [19,34,49,63,67,84,115]. For example, the “I Immunize” campaign trained parents as vaccine advocates to mobilize their communities to be more informed about the importance of childhood vaccines [69]. Lastly, recall and reminder systems for parents and HCP have been recommended for encouraging one-on-one discussion about childhood vaccinations [33,97].

#### 3.4.10. Technology-Based Communication Strategies

Technology use in the U.S. is now widespread, with parents using the Internet and social media via computers and cellphones to search for health information [116]. As a result, information about vaccination in particular has changed in origin, nature, and speed. The Internet and Web 2.0 now produce, relay, and significantly accelerate the spread of vaccine information. While the Internet has allowed for free and anonymously accessible information, which can empower parents to own their health-related decisions, it has come with extreme costs. Information on the Internet is uncontrolled, and searches undertaken by web users are heavily dictated by algorithms, which contribute to hearsay and even magnify and amplify it [15]. Anti-vaccination messages, in particular, have largely been successful on the Internet and social media platforms, utilizing message-framing techniques such as storytelling and fear-based messaging to grab the audience’s attention and influence the number of shares [19,23,38,57,76,87,93]. A closer look at the successes of the anti-vaccination messages has, in turn, led to strategic pro-vaccine communication and social marketing interventions and campaigns on the Internet and Web 2.0 to improve positive vaccine beliefs, attitudes, intentions, and behaviors. In two randomized controlled trials (RCTs) that utilized a web-based social media intervention, parents were referred to vaccination information on a website or on a social media platform or were given standard paper-based vaccination information from their HCP. In both RCTs, it was found that referring parents to vaccine information on websites with social media components was more effective at increasing positive vaccination attitudes than distributing standard paper-based vaccination information [57,70]. Mobile applications on smartphones and tablets have also been shown to positively reinforce positive vaccination attitudes, behaviors, and intentions to vaccinate [74,100,105]. However, it is noted in several studies that for interventions interested in utilizing the Internet and social media to deliver vaccination information, it is critical to understand that this technology rapidly evolves, and therefore, these interventions need to be monitored and updated constantly [14,26,61,70].

## 4. Discussion

The main objective of this systematic literature review was to explore key health communication strategies and interventions that could be successful in addressing and preventing parental vaccine hesitancy towards childhood vaccines in the long term. This systematic literature review attempted to reduce reviewer bias through the use of objective reproducible criteria to select relevant publications and articles and assess their validity. One of the main strengths of the systematic literature review includes a narrow focus on the research question and objective. Other strengths include the comprehensive search for evidence, the criteria-based selection of relevant evidence, the analysis for validity, the objective summary of themes identified from the literature, and the evidence-based inferences. However, the strengths of a systematic literature review can also be weaknesses. The narrow focus of this systematic literature review did not allow for comprehensive coverage of vaccine hesitancy for all age groups, vaccines, and geographic contexts. Furthermore, only articles in English were assessed, not accounting for other publications and articles written in multiple languages. Finally, while the systematic literature review attempted to reduce review bias in the methodology, the evidence-based inferences from the study results could produce some bias.

### 4.1. Public Health Implications of the Findings

To date, there has not been a recent review on the trends and evidence base around health communication interventions and strategies to address parental vaccine hesitancy. In order to address this growing phenomenon, it is imperative that we understand which strategies are proving effective and which strategies are proving ineffective. Furthermore, there needs to be a stronger focus on preventing vaccine hesitancy.

### 4.2. Interventions That Have Been Successful or Unsuccessful

This systematic literature review provided an overview of interventions that were identified in the literature in the time period of January 2008 and October 2019. No single intervention or strategy was deemed to be effective on its own in addressing the complexity of parental vaccine hesitancy. Overall, the interventions that demonstrated the most success were multi-component, employed a variety of media or touchpoints, incorporated some dialogue element, and were personalized and tailored to the target populations’ specific vaccine concerns, historical experiences, religious or political affiliations, socioeconomic status, and trusted messengers for information. The specific components and communication strategies of interventions that proved essential for their success included the use of messaging-framing techniques such as storytelling with the use of gists, emotive anecdotes, and imagery. In addition, message-framing techniques that utilized science-based messaging also proved to be persuasive if presented in plain and simple language with the avoidance of medical and clinical jargon. Specific components and communication strategies that proved unsuccessful and sometimes detrimental included the use of message-framing techniques that used heavy statistical and numerical messaging, fear-based messaging, and messaging that attempted to correct or debunk commonly held myths and misconceptions about childhood vaccination. Several studies have proposed that future research should address the challenges of combating misinformation and disinformation. Furthermore, studies have pointed to the need to explore monitoring and evaluation systems for identifying vaccine-hesitant parents and strategically targeting interventions. Lastly, arguments have been made to mimic the effective strategies used in anti-vaccination movements, from the message-framing techniques to their tailoring of messaging and use of messengers for campaigns.

## 5. Recommendations and Conclusions

The following recommendations are intended to aid public health professionals in adapting, implementing, and scaling up health communication interventions and strategies to address and prevent parental vaccine hesitancy towards childhood vaccines in the long term. These recommendations are based on the catalogue of interventions and themes identified in the literature.

### 5.1. Start Early and Build Trust with Parents

Interventions should target parents early in pregnancy and target first-time mothers and parents, taking advantage of prenatal appointments and the first postnatal appointment. This is the time when parents are thinking about childhood immunizations the most and deciding whether to accept the recommended schedule, be selective about immunization, or outright refuse immunizations for their child.

### 5.2. Tailor the Information to the Target Audience, Their Reasons for Hesitancy, and the Specific Context

Parents want vaccine information that is tailored to their needs and tailored to their child. Future interventions should ensure that the needs of the target population are well studied. This includes understanding the target population’s specific vaccine concerns, historical experiences, religious or political affiliation, socioeconomic status, demographic background, and trusted messengers for information. This also includes targeting interventions depending on where parents reside on the vaccine-hesitancy continuum scale: those who accept all vaccines but are still concerned, those who refuse or delay some vaccines but accept others, and those who refuse all vaccines. Furthermore, vaccine messaging and stimuli should evoke different values. The message and messenger of vaccine information must be able to appeal to the values of the target audience in order to change vaccine behaviors.

### 5.3. Present Vaccination as the Default Approach

Health care providers need to be better equipped to present vaccination as the default approach and to address the issues of vaccine hesitancy through more training. Another solution could be the introduction of a “triage and treat” approach to clinical assessment and care for vaccine-hesitant parents. With regard to vaccine messaging, there needs to be a strong focus on presenting vaccination as the social norm. The challenge of today’s media environment is the amplification of dissenting views and perspectives. It inaccurately portrays that a larger proportion of individuals are hesitant or refusing vaccination.

### 5.4. Develop Vaccine Education Materials Using Health Literacy Best Practices

Parents that are on the vaccine-hesitancy scale often request science-based vaccine messaging in plain language that avoids medical jargon. Vaccine education materials should therefore go through an evaluation process before being released to the public. This does not mean that materials need to be “dumbed down”; rather, materials need to be evaluated to ensure the content, word choice and style, use of numbers, organization, layout and design, and use of visual aids is appropriate for the target audience.

### 5.5. Incorporate Dialogue-Based Communication Strategies and Provide Balanced Information about Vaccines

Interventions that incorporate dialogue elements are the most effective. Parents want an interactive and engaging approach to learning about childhood vaccines. Furthermore, parents want balanced information about both the benefits and risks including the side effects of vaccines. Parents report that too often, the standard childhood vaccination messaging focuses exclusively on the benefits and does not address the possible risks and side effects, which leads parents to have more questions and more concerns.

### 5.6. Tell Stories with Gists and Emotive Anecdotes

The effectiveness of vaccine communication strategies depends on the message-framing techniques used. Messaging in the form of storytelling with the use of gists, emotive anecdotes, and imagery has been shown to be among the most persuasive messaging strategies. The use of fear-based messaging should be avoided, as it has proven to reinforce vaccine hesitancy in parents.

### 5.7. Choose the Messenger of Vaccine Information Carefully

Health care providers are identified as the most trustworthy messengers of vaccine information. Future interventions involving HCP should carefully consider and mitigate the potential for any unintended consequences, i.e., disturbance of the trust between HCP and parents. A second important messenger of vaccination information is a parent’s social network, which can include family, friends, colleagues, neighbors, and other personal relations. It has been found that a parent’s social network can be more predictive of a parent’s decision to vaccinate their child than any other variable, including the parent’s own perceptions of vaccination.

### 5.8. Focus on Vaccine Messaging That Centers Both the Child and Community

It is unclear whether it is more advantageous to frame vaccine messaging as a social responsibility or an individual’s responsibility. Therefore, it is important to provide both kinds of messaging and tailor the message specifically to the target population’s background and values.

### 5.9. Use Technology to Promote Vaccination

Internet-based interventions to address parents’ vaccine concerns can be a constructive strategy since parents often seek vaccine information on the Internet. Utilizing web-based platforms with social media applications has shown a lot of promise in engaging this target group. Furthermore, new mothers and parents often use baby apps and other phone apps throughout their pregnancy and after childbirth for information on how to keep their child healthy. It could be advantageous to develop a phone app for new moms/new parents with information about childhood vaccines.

### 5.10. Use Caution When Addressing Vaccine Misinformation

Several emerging studies have shown a backfire effect when interventions attempt to correct or debunk vaccine misinformation, leading to the intervention actually reinforcing strongly held false beliefs about vaccines and increasing vaccine hesitancy. This is an area for further research. What is clear, however, is that there must be improvements in the monitoring and tracking of the spread of misleading and inaccurate vaccine information on the Internet. If misinformation cannot be corrected, it needs to be prevented.

### 5.11. Improve Parents’ Vaccine Literacy and Critical Thinking Skills

The expansion of the Internet has allowed for vaccine misinformation and fear-based anti-vaccine messages to have a greater reach and impact. There is a need for interventions to combat the spread of misinformation and disinformation. There is an enormous need for interventions to focus on improving vaccine literacy and critical thinking for parents. This could involve a series of community interventions that introduce parents to how to be an informed health care consumer and, more specifically, a more informed consumer of vaccine information.

## Figures and Tables

**Figure 1 vaccines-08-00590-f001:**
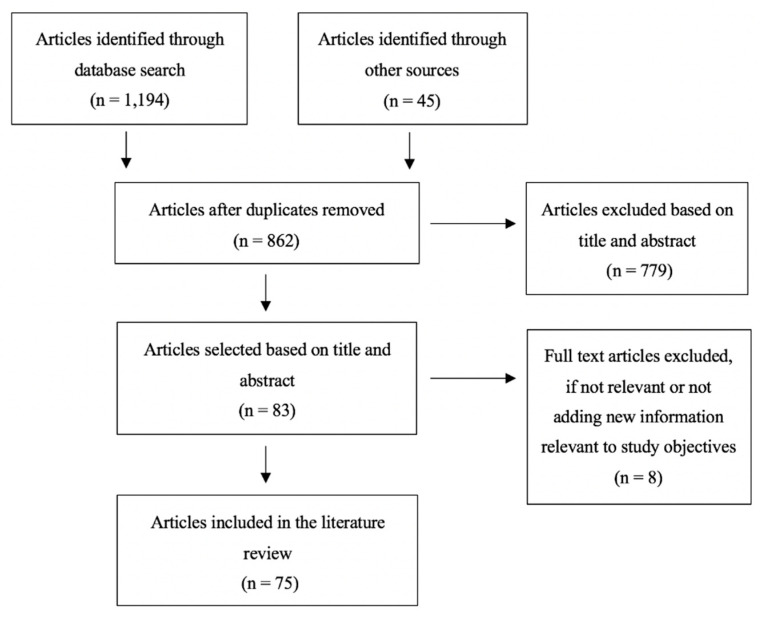
Preferred Reporting Items for Systematic Reviews and Meta-Analysis (PRISMA) flow diagram.

**Figure 2 vaccines-08-00590-f002:**
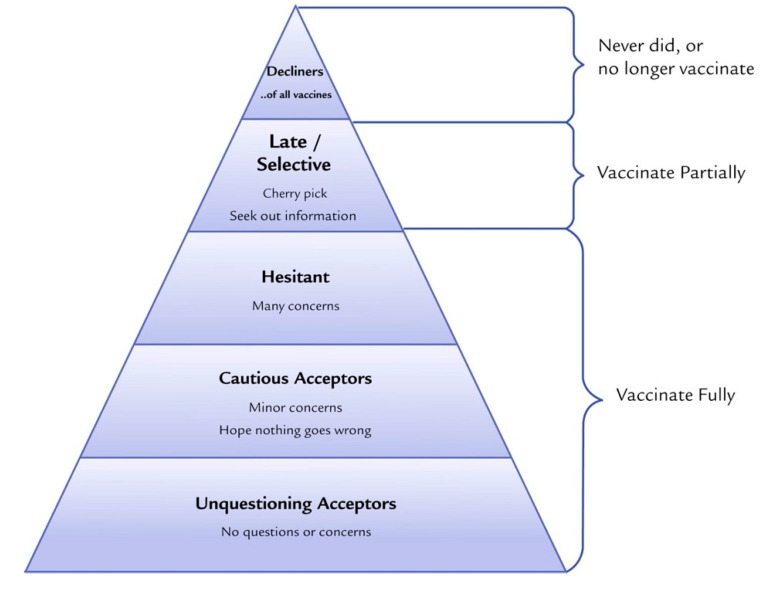
Vaccine acceptance spectrum from Leask et al. 2015. Improving Communication about Vaccination, adapted from Julie Leask’s online blog [60].

**Table 1 vaccines-08-00590-t001:** Key components of approved vaccines in the U.S. reproduced from the CDC [46].

Type of Ingredient	Example(s)	Purpose
Preservatives	Thimerosal (only in multi-dose vials of flu vaccine)	To prevent contamination
Adjuvants	Aluminum salts *	To help boost the body’s response to the vaccine
Stabilizers	Sugars, gelatin	To keep the vaccine effective after manufacture
Residual cell culture materials	Egg protein	To grow enough of the virus or bacteria to make the vaccine
Residual inactivating ingredients	Formaldehyde	To kill viruses or inactivate toxins during the manufacturing process
Residual antibiotics	Neomycin	To prevent contamination by bacteria during the vaccine manufacturing process

* Aluminum salts have been the only adjuvant used in most FDA-approved and licensed vaccines in the U.S.A.; other adjuvants such as ASO4, ASO1, etc. have also received FDA approval during the past decade, but not for vaccines used in children currently.

**Table 2 vaccines-08-00590-t002:** Recommended vaccinations for infants and children (birth through 6 years) reproduced from the CDC [48].

Birth	1 Month	2 Months	4 Months	6 Months	12 Months	15 Months	18 Months	19–23 Months	2–3 Years	4–6 Years
HepB	HepB		-	HepB				-	-	-
-	-	RV	RV	RV	-	-	-	-	-	-
-	-	DTaP	DTaP	DTaP	-	DTaP		-	-	DTaP
-	-	Hib	Hib	Hib	Hib		-	-	-	-
-	-	PCV13	PCV13	PCV13	PCV13		-	-	-	-
-	-	IPV	IPV	IPV				-	-	IPV
-	-	-	-	Influenza (Yearly) *						
-	-	-	-	-	MMR	-	-	-	-	MMR
-	-	-	-	-	Varicella	-	-	-	-	Varicella
-	-	-	-	-	HepA				-	-

* Two doses given at least four weeks apart are recommended for children age 6 months through 8 years of age who are getting an influenza (flu) vaccine for the first time and for some other children in this age group.

**Table 3 vaccines-08-00590-t003:** Table of search terms used.

Target Population	Children and Parents
Vaccine hesitancy	Parental vaccine concerns, vaccine refusal and delay, childhood vaccines, alternative vaccine schedules
Determinants of vaccine hesitancy	Vaccine attitudes, vaccine behaviors, vaccine decisions vaccine risk perceptions
Addressing vaccine hesitancy	Vaccine confidence, vaccine acceptability, vaccine promotion, vaccine communication, addressing/preventing vaccine hesitancy, intervention strategies, pro-vaccine messages

**Table 4 vaccines-08-00590-t004:** The WHO SAGE Working Group Determinants of Vaccine Hesitancy Matrix [29].

CONTEXTUAL INFLUENCES Influences arising due to historic, socio-cultural, environmental, health system/institutional, economic, or political factors	(a) Communication and media environment (b) Influential leaders, immunization program gatekeepers, and (c) anti- or pro-vaccination lobbies (d) Historical influences (e) Religion/culture/gender/socio-economic factors (f) Politics/policies (g) Geographic barriers (h) Perception of the pharmaceutical industry
INDIVIDUAL AND GROUP INFLUENCES Influences arising from personal perception of the vaccine or influences of the social/peer environment	(a) Personal, family, and/or community members’ experience with vaccination, including pain (b) Beliefs, attitudes about health and prevention (c) Knowledge/awareness (d) Health system and providers—trust and personal experience (e) Risk/benefit (perceived, heuristic) (f) Immunization as a social norm vs. not needed/harmful
VACCINE/VACCINATION-SPECIFIC ISSUES Directly related to vaccine or vaccination	(a) Risk/benefit (epidemiological and scientific evidence) (b) Introduction of a new vaccine or new formulation or a new (c) recommendation for an existing vaccine (d) Mode of administration (e) Design of vaccination program/mode of delivery (e.g., (f) routine program or mass vaccination campaign) (g) Reliability and/or source of supply of vaccine and/or (h) vaccination equipment (i) Vaccination schedule (j) Costs (k) The strength of the recommendation and/or knowledge base (l) and/or attitude of health care professionals

**Table 5 vaccines-08-00590-t005:** Messengers of vaccination information.

Messenger	Evidence from the Literature
Health Care Professionals	Health care professionals (HCP) are cited as the most important source for receiving vaccination information for parents. They are seen as trustworthy, informative, and a reliable source for answering questions and concerns parents have about childhood vaccines. HCP’s communication styles with parents are important. Poor communication and negative relationships with HCP can heavily impact parents’ vaccination decisions. Furthermore, HCP’s behavior and opinions about vaccination influence parents’ acceptance of vaccination. HCP may not always be best equipped or have the time to address the emerging issue of parental vaccine hesitancy. This is especially true when parental questions and concerns arise from very persuasive misinformation found on the Internet and from their social network, which has proven difficult to counter and correct based on numerous studies. The literature suggests that if HCP refuse parents’ requests to delay vaccines, be selective with vaccines, or alter the recommended childhood vaccine schedule, parents will continue to search for other HCP or alternative health professionals. However, literature also suggests that if vaccination is presented as the default approach and HCP engage in open discussion with parents about vaccines early and often, parents are more likely to vaccinate their children [4,14,15,23,25,27,34,58,59,61,63,73,77,78,79,80,81,82,83,84,85,86].
Internet and Social Media (Web 2.0)	Parents often cite that the Internet and social media are trusted sources of vaccination information. In one study, up to 72% of American internet users trusted health information obtained on the Internet, and 75% evaluated the source of information only sometimes or never. This poses a tremendous threat and opportunity for individuals to be influenced by false or misleading information about vaccines [38,59,70,76,80,87,88,89,90].
Family and Friends	A parent’s social network plays an important role in their attitudes and beliefs about vaccines and their intention to vaccinate their child. A parent’s social network can include family, friends, colleagues, neighbors, and other personal relationships. It is found in the literature that a parent’s social network can even be more predictive of a parent’s decision to vaccinate their child than any other variable, including the parent’s own perceptions of vaccination. When vaccination is viewed as the social norm in that parent’s social network, social pressure and responsibility act as a powerful driver of vaccine uptake [4,15,59,61,76,91,92,93,94].
Religious and Community Leaders	Religious and community leaders have proven to be effective messengers for motivating parents to vaccinate or not vaccinate their child. Vaccine refusal has been linked to moral convictions and philosophical beliefs such as a preference for natural over artificial medicines. Therefore, for specific populations, it may be important to identify religious and community influences in order to successfully deliver vaccination information and tailor specific interventions [4,19,27,75,95].
Health Authorities and Government Authorities	Health authorities and government authorities are cited as one of the most common barriers to vaccine acceptance for some vaccine-hesitant populations. This is due to a lack of trust in the messenger due to many reasons including political, social, historical, etc. However, during outbreaks of vaccine-preventable diseases, traditional media and government websites are often used, despite the widely held belief that social media is replacing these legacy news organizations/authorities [4,19,24,26,82,86,93].

**Table 6 vaccines-08-00590-t006:** Content of vaccination information.

Vaccine Topic	Evidence from the Literature
Science of Vaccines	Parents want up-to-date information on current scientific information, research, and statistics related to past and current vaccines [35,63,77,82,89,101].
Vaccine Safety	Parents want to be informed about the safety of vaccines. This includes information about common vaccine side-effects and their severity. Furthermore, parents want vaccine information to be balanced; they want information about both the benefits and the risks of vaccines. Parents report that they are overwhelmingly informed about the benefits and under-informed about the risks, which leads them to be skeptical. Parents also want information about how to mitigate the pain that is inflicted on their child when receiving a vaccine. There are clinically based guidelines that have been developed to reduce vaccination-associated pain. Parents want to be educated about pain management for vaccine injection before or on the day of vaccination [14,19,25,26,27,32,77,81,82,93,102].
Vaccine Ingredients	Parents want information about the ingredients in vaccines. In particular, they want to know the purpose of each ingredient and if those ingredients are toxic and unsafe [23,26,32,77,79].
Combined versus Single Vaccines	Parents want information about vaccine dosage and the differences between combined and single vaccines [25,26,27,65,77].
The Childhood Vaccine Schedule	Parents want information about the reasons behind the childhood vaccine schedule: clarification on the quantity and timing of vaccines [25,27,73,75,76,77,85,103].
The Diseases that Vaccines Prevent	Parents may lack knowledge about how vaccinations work and the diseases they prevent. Today’s generation of parents may not have first-hand knowledge of once-deadly infectious diseases that are now prevented by vaccines [19,23,65,77,79].
Technical Information on Vaccine Production and Delivery	Parents want more information about the names of vaccines, how vaccines have been tested, the proper storage of vaccines, the country of manufacture, and the quality control/evaluation measures for production and delivery [35,77].
Vaccine Policies, Recommendations, and Costs	Parents want to know the reasons behind why some policies and recommendations differ for different communities, school districts, etc. [35,77].
Alternatives to Vaccines	Some parents are exploring alternatives to vaccines such as homeopathic treatments. Parents want information about the effectiveness and safety of vaccines compared to other alternatives [4,35,49,76,85,89,93].
Myths about Vaccines	Parents are concerned when faced with myths and controversies about childhood vaccines. Examples of major vaccine myths circulating today include the ideas that vaccines cause autism, and mercury in vaccines act as a neurotoxin. Parents want clarification about why these myths exist and scientific evidence and expert advice on how to recognize vaccine misinformation [18,26,82,89,93,97].

**Table 7 vaccines-08-00590-t007:** Message-framing techniques for vaccination information.

Message-Framing Technique	Evidence from the Literature
Storytelling	Storytelling and the use of personal narratives when communicating vaccination information is a powerful messaging tool. It has more famously been used in anti-vaccination messaging to spread fear about childhood vaccinations. Two examples of pro-vaccine studies that have utilized storytelling include the Seattle Mama Doc and Moms Who Vax [25,27,38,63,78,80,89,93,101,104,105].
Gists	Gists or the bottom-line meaning of messaging is critical in storytelling messaging. Stories without gists may not be as effective. Examples of gists used in vaccine messaging include “there is no chance that mercury in vaccines can cause autism, since it is not in vaccines anymore” or “if you do not vaccinate your child, there is a real chance that they could get sick”. Messages about vaccines that express a clear gist are arguably more compelling compared to messages without gists. In general, gists are also more likely to be shared on Facebook and other social media platforms than verbatim statistics [38,63,87,90,97,102,105,106].
Emotive Anecdotes and Emotive Imagery	Emotive anecdotes and emotive imagery in messaging are cited as one of the most persuasive and effective message-framing strategies to communicate vaccine information. An anecdote is a short amusing or interesting story about a real incident or person that has a point to make. The emotive anecdote is the use of emotive language and images to have a greater emotional impact on the audience [18,24,38,79,80,84,86,87,89,90,100,101,102].
Gamified Messaging	Offering parents a gamified learning experience can significantly contribute to knowledge gain in the context of vaccination. This is a new and developing message-framing technique being used in this field with promising initial results [100].
Tone	Compared to more serious messaging, satirical messaging and humor has been shown to reduce psychological reactance, leading to greater perceptions of disease severity and less vaccine hesitancy. While humor can reduce psychological reactance, it is not an effective strategy among parents who already hold favorable beliefs about vaccination. Among parents who are not vaccine hesitant, the use of humor in messaging is less effective and can even be counter effective in positively influencing parents’ attitudes, beliefs, and intentions towards childhood vaccinations. In addition to the use of humor, the use of enthusiasm in the messenger’s tone when delivering vaccine communication is often received more positively among parents compared to unenthusiastic communication. This has been shown in the motivational interviewing techniques used in interventions aimed at new mothers [71,81,107,108].
Gain versus Loss Goal Framing	Gain versus loss goal framing is well studied in the literature as an important health communication message-framing technique. For vaccination messaging, the hypothesis is that loss-framed messages that present the negative consequences of not performing a behavior are more persuasive for disease-detection behaviors, and gain-framed messages, which present the positive consequences of performing a behavior, are more persuasive for preventive behaviors. It is argued that because vaccination may be considered risky given the possibility of side effects, loss-framed messages might be more effective. There is mixed evidence that loss-framed messaging is more effective than gain-framed messaging for vaccination information. Some studies affirm that loss-framed messages have the greatest impact, other studies find that a combination of gain- and loss-framed messaging is most effective, and other studies find no overall difference between loss- and gain-framed messaging [27,35,36,67,108,109,110].
Fear-Based Messaging	Fear-based messaging is more often than not viewed as counterproductive in reducing vaccine hesitancy by several studies. By contrast, fear-based messaging has been extremely effective in anti-vaccination messaging [18,24,80,86,93,97,111].
Co-Promoted Behavioral Messaging	Co-promoted behavioral messaging is effective in framing vaccines as part of a holistic health plan for the child. Pairing pro-vaccine messages with messages such as breastfeeding and healthy eating has been shown to be effective. Furthermore, research has shown that it is advantageous to appeal to the target audiences’ values in order to change behaviors. Vaccine decisions are value-based decisions, and the goal should be not to change values but to change behaviors [69,89,91,96,102,112].
Statistical Information	Statistical information and (“verbatim”) statements on probabilities about vaccines are not as effective on their own and are not found to be as powerful as using anecdotes and emotion-based messaging techniques. The use of statistical information and probabilistic information is most often used by medical and public health authorities in their communication about vaccines. This message-framing technique is also more commonly used in the knowledge-deficit approach to address vaccine hesitancy [4,18,24,38,63,86,87,93,101,104].
Science-Based Messaging	Messaging with references to scientific studies and references to the science of vaccines is cited with a high level of support from parents. However, parents have reported high levels of support for science-based messaging but not when science-based jargon is used. Parents want to be able to understand the science of vaccines in plain language [35,82,89,101].
Expert Opinion	Expert opinion (i.e., an article written by a “doctor” or “scientist”) messaging has been shown to be powerful. Some interventions show that messaging including quotes from doctors and individuals seen as experts in the field of vaccines is very persuasive [89,102].

**Table 8 vaccines-08-00590-t008:** Common misconceptions about childhood vaccination.

Misconceptions about Childhood Vaccination	Evidence from the Literature
Natural Immunity is Better than Vaccine-Acquired Immunity	Some vaccine-hesitant parents claim that natural immunity is better than immunity induced by vaccines, which they believe are “toxins”. Other vaccine-hesitant parents claim that most vaccine-preventable diseases are harmless to most children in the U.S. today, and natural exposure provides more long-lasting immunity [19,22,25,76,84,95,105].
Too Many Immunizations Overload the Immune System	Some vaccine-hesitant parents claim that vaccines “overwhelm” the immune system and that the childhood vaccine schedule involves too many vaccines, too soon. There is a belief that in following the recommended vaccine schedule for children, the immune system becomes “overwhelmed” and this leads to autism and an assortment of chronic health conditions [19,22,25,49,65,79,84,85,91,105].
Vaccines Contain Harmful Ingredients and Cause Serious Side Effects	Some vaccine-hesitant parents often state that vaccines are more dangerous than the infectious diseases that vaccines prevent. Vaccine-hesitant parents believe strongly that the toxins in vaccines can lead to an assortment of chronic health conditions that are more dangerous than vaccine-preventable diseases such as measles, mumps, and rubella (MMR) [23,25,26,77].
Many of the Vaccine-Preventable Diseases Are Uncommon in the United States Now	Until recently with the 2020 COVID-19 pandemic in the U.S., most parents today never had first-hand experience with infectious disease outbreaks. Therefore, it has been cited that either the side effects of the vaccines themselves appear more concerning than the diseases that the vaccines prevent, or parents believe vaccines are not needed [19,23,85].
There Are Alternatives to Vaccines	Some vaccine-hesitant parents believe that there are alternative to vaccines that are more effective and safer than vaccines for protecting children’s health. Vaccine-hesitant parents may seek counseling and information from alternative medicine practitioners such as homeopaths and naturopaths to justify alternative to vaccines [49,69,76,85,91].
Pharmaceutical Companies and Medical Science are not Trustworthy	Some vaccine-hesitant parents believe that pharmaceutical companies are using vaccines as a tool to profit without consideration of the harm to children. Some parents also have distrust in the medical science behind vaccines and believe that the research on vaccines is outdated and inaccurate and vaccines have not been tested properly and therefore are not safe [22,24,27,82,86].
Vaccines Cause Autism	A significant number of vaccine-hesitant parents believe, or are uncertain, about the scientifically unproven link between vaccines and autism. [18,26,82,89,93,97].

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
