# Peer review of "Addressing Parental Vaccine Hesitancy towards Childhood Vaccines in the United States: A Systematic Literature Review of Communication Interventions and Strategies"

_vaccines, 2020, doi:10.3390/vaccines8040590_

Round 1

Reviewer 1 Report

The authors present a thorough literature review of parental vaccine hesitancy concerning childhood vaccines in the US, defined by the authors as vaccines used in infants and children less than 6 years of age. The review aims to address three objectives: (1) definition of vaccine hesitancy; (2) identify key determinants of vaccine hesitancy for childhood immunizations, and (3) explore key health communication interventions and strategies that could be successful in addressing and preventing parental vaccine hesitancy towards childhood vaccines in the long term. However, the authors also state that the primary focus of the review is the 3rd objective. Indeed, there is virtually no discussion of vaccine hesitancy. Although the authors state that there are several definitions (l. 216), only the WHO definition is provided and the authors seem to accept this definition. Similarly, under “Key determinants of childhood vaccine hesitancy”, the authors simply present the Vaccine Hesitancy Determinants Matrix and do not discuss whether, in their opinion, these terms are correct or whether any determinants have been excluded. It is very useful and appropriate to provide the WHO definition and the determinants, but I would suggest that the authors not present these as objectives of the literature review. Again, the main focus is the communication interventions and strategies, and the authors provide a nice summary of recommendations.

Other comments:

  1. 48 – “there continues to be an increase in the number of children with no vaccines”. Please provide the years over which this increase was determined.
  2. 67-68 – “….no clear scientific evidence….”. Please remove “clear”. There is NO scientific evidence that vaccine cause autism.
  3. 117 – “specific humoral (antibodies and cytokines)”. Please remove “and cytokines”. While cytokines are soluble factors, they are not considered part of the humoral immune response and play an equally important role in the cell-mediated immune response (and are not antigen-specific).
  4. 118 – 120. “Vaccines can be further divided into four main categories: live attenuated vaccines, killed vaccines, subunit vaccines and nucleic acid (DNA and mRNA) vaccines [39]”. Another category is vectored vaccines, e.g. the adenovirus-vector vaccines currently pursued for COVID-19. Current childhood vaccines only fall into three categories, attenuated, killed and subunit. In addition, reference 39 does not seem appropriate. There are better reviews of vaccines.
  5. 122 – Ref. 43 is included twice.
  6. 128 – Components of vaccines
  7. 135 – “…..prime innate immune system…”. Change to “..activate the innate immune system…”
  8. 138 – “…protect the vaccine from contamination with external contaminations”. Change to protect the vaccine from external contaminations”
  9. Table 1 lists aluminum salts as the only vaccine adjuvant. In the past 10 years or so, the FDA has approved vaccines with other adjuvants such as ASO4, ASO1, etc. Vaccines with these adjuvants are not for use in children currently.
  10. Table 2 and Figure 1 are presented as “adapted from the CDC”, but in fact are identical to the information presented on the CDC website. I don’t think that Table 2 is necessary for the purpose of this paper, and the authors can simply refer to the CDC website. What is the meaning of the yellow boxes in Figure 1?
  11. 162 – Please write out human papilloma virus (HPV) in full.
  12. 178 – 179. “International publications from a few high-income, English-dominant countries including the U.K., Australia and Canada were also included.” This suggests that the remaining articles were exclusively focused on the US? Please state if this is indeed the case.
  13. 190 – “Articles were excluded if they focused primarily on LMICs”. If the literature was focused exclusively on the US with exception of a few articles from a few high-income countries, then this statement is not needed.
  14. 222 – 224. This sentence repeats verbatim the sentence in the introduction. Please refer to the comments above about the objectives.
  15. 316 – Change important to importance.
  16. 328 – How short is a “short term benefit”?
  17. 361-362. “parents are receptive to the message only if they find the messenger trustworthy and credible.” The authors may want to comment on the lower vaccination rate in underrepresented minorities and whether physicians and health care workers who look like the parents may be more effective in communicating the need for vaccination.
  18. Table 6 under Tone: “Compared to more serious messaging, satirical messaging and humor has been shown to reduce psychological reactance and lead to a greater perception of disease severity which vaccines prevent among individuals who already endorse false beliefs about childhood vaccines.” I don’t understand this sentence – please rephrase.

Author Response

RESPONSE TO REVIEWER COMMENTS

REVIEWER 1

Comments and Suggestions for Authors

The authors present a thorough literature review of parental vaccine hesitancy concerning childhood vaccines in the US, defined by the authors as vaccines used in infants and children less than 6 years of age. The review aims to address three objectives: (1) definition of vaccine hesitancy; (2) identify key determinants of vaccine hesitancy for childhood immunizations, and (3) explore key health communication interventions and strategies that could be successful in addressing and preventing parental vaccine hesitancy towards childhood vaccines in the long term.

However, the authors also state that the primary focus of the review is the 3rd objective. Indeed, there is virtually no discussion of vaccine hesitancy. Although the authors state that there are several definitions (l. 216), only the WHO definition is provided and the authors seem to accept this definition. Similarly, under “Key determinants of childhood vaccine hesitancy”, the authors simply present the Vaccine Hesitancy Determinants Matrix and do not discuss whether, in their opinion, these terms are correct or whether any determinants have been excluded. It is very useful and appropriate to provide the WHO definition and the determinants, but I would suggest that the authors not present these as objectives of the literature review. Again, the main focus is the communication interventions and strategies, and the authors provide a nice summary of recommendations.

Response: We thank the reviewer for a thorough review and extremely relevant points. We have modified the sentence and description for better clarity.

Other comments:

  1. 48 – “there continues to be an increase in the number of children with no vaccines”. Please provide the years over which this increase was determined.
  2.  

Response: Dates are now provided

  1. 67-68 – “….no clear scientific evidence….”. Please remove “clear”. There is NO scientific evidence that vaccine cause autism.

Response:  Clear word has been deleted.

  1. 117 – “specific humoral (antibodies and cytokines)”. Please remove “and cytokines”. While cytokines are soluble factors, they are not considered part of the humoral immune response and play an equally important role in the cell-mediated immune response (and are not antigen-specific).

Response:  We have accepted reviewer’s suggestion and removed cytokines. However, cytokines, complement components, anti-microbial peptides are some of the humoral factors participating in different aspects of immunity. Cytokines, in particular do help activation of Tfh and antibody production, among other effects.

  1. 118 – 120. “Vaccines can be further divided into four main categories: live attenuated vaccines, killed vaccines, subunit vaccines and nucleic acid (DNA and mRNA) vaccines [39]”. Another category is vectored vaccines, e.g. the adenovirus-vector vaccines currently pursued for COVID-19. Current childhood vaccines only fall into three categories, attenuated, killed and subunit. In addition, reference 39 does not seem appropriate. There are better reviews of vaccines.

Response:  We have added ‘Vectored vaccines’ which inadvertently got left out originally.

  1. 122 – Ref. 43 is included twice.

Response:  Corrected

  1. 128 – Components of vaccines
  2. 135 – “…..prime innate immune system…”. Change to “..activate the innate immune system…”

Response:  Changed as suggested

  1. 138 – “…protect the vaccine from contamination with external contaminations”. Change to protect the vaccine from external contaminations”

Response:  Changed as suggested

  1. Table 1 lists aluminum salts as the only vaccine adjuvant. In the past 10 years or so, the FDA has approved vaccines with other adjuvants such as ASO4, ASO1, etc. Vaccines with these adjuvants are not for use in children currently.

Response:  We have added a footnote to reflect the valuable reviewer suggestion. Thanks.

  1. Table 2 and Figure 1 are presented as “adapted from the CDC”, but in fact are identical to the information presented on the CDC website. I don’t think that Table 2 is necessary for the purpose of this paper, and the authors can simply refer to the CDC website. What is the meaning of the yellow boxes in Figure 1?

Response:  We totally agree with the reviewer and previous table 2 has now been deleted and the readers are advised to check out the CDC website.

  1. 162 – Please write out human papilloma virus (HPV) in full.

Response:  Done as suggested.

  1. 178 – 179. “International publications from a few high-income, English-dominant countries including the U.K., Australia and Canada were also included.” This suggests that the remaining articles were exclusively focused on the US? Please state if this is indeed the case.

Response:  The search methods and details are edited to clarify the raised point.

  1. 190 – “Articles were excluded if they focused primarily on LMICs”. If the literature was focused exclusively on the US with exception of a few articles from a few high-income countries, then this statement is not needed.

Response:  The search methods and details are edited to clarify the raised point.

  1. 222 – 224. This sentence repeats verbatim the sentence in the introduction. Please refer to the comments above about the objectives.

Response:  Addressed as modified under the objectives.

  1. 316 – Change important to importance.

Response:  Changed as suggested.

  1. 328 – How short is a “short term benefit”?

Response:  Modified to provide sought detail.

  1. 361-362. “parents are receptive to the message only if they find the messenger trustworthy and credible.” The authors may want to comment on the lower vaccination rate in underrepresented minorities and whether physicians and health care workers who look like the parents may be more effective in communicating the need for vaccination.

Response:  We are assuming the reviewer is asking for a comment that reflects racial and ethnic background of the messenger. We have added a sentence: “It has also been suggested that racial and/or ethnic background of the messenger (health care worker and physician) may also promote effective communication to improve vaccination rates in underrepresented minorities.” I hope this is acceptable.

Table 6 under Tone: “Compared to more serious messaging, satirical messaging and humor has been shown to reduce psychological reactance and lead to a greater perception of disease severity which vaccines prevent among individuals who already endorse false beliefs about childhood vaccines.” I don’t understand this sentence – please rephrase.

Response:  We have modified it for better clarity.

Reviewer 2 Report

The manuscript by Olson et al., is a very interesting review about the concept of Vaccine Hesitancy. The subject is very important. It’s currently something of a “hot topic”. Given the complexity involved, the authors have produced many positive and welcome outcomes. The literature review offers a useful overview of current research and policy, and the resulting bibliography provides a very useful resource for current practitioners

Overall, this research is well written, and the content of this manuscript is of major interest. I found this manuscript well-balanced.

This manuscript deserves to be published in the pages of the Vaccines. Only two minor changes are necessary:

Line 9: Please correct “Correspondence: Corresponding authors”

Tables: They are very large and need to be formatted adequately. There are too many unused spaces and you can use a smaller font for the text

Author Response

REVIEWER 2

Comments and Suggestions for Authors

The manuscript by Olson et al., is a very interesting review about the concept of Vaccine Hesitancy. The subject is very important. It’s currently something of a “hot topic”. Given the complexity involved, the authors have produced many positive and welcome outcomes. The literature review offers a useful overview of current research and policy, and the resulting bibliography provides a very useful resource for current practitioners. Overall, this research is well written, and the content of this manuscript is of major interest. I found this manuscript well-balanced. This manuscript deserves to be published in the pages of the Vaccines.

Response:  We thank the reviewer for positive and favorable comments.

Only two minor changes are necessary:

Line 9: Please correct “Correspondence: Corresponding authors”

Response:  Modified as per the journal style.

Tables: They are very large and need to be formatted adequately. There are too many unused spaces and you can use a smaller font for the text.

Response:  We have reformatted some of the larger tables to maximize space utilization.

Round 2

Reviewer 1 Report

The authors have adequately addressed my comments. A few minor edits remain:

Line 48 - change "has" to "have"

Line 144 - remove "experimental". These adjuvants are now used in licensed vaccines and have passed the preclinical and clinical development stage.

Line 562 - correct to "anecdotes"

Author Response

We thank the reviewer for the following corrections / comments.

Line 48 - change "has" to "have"

Response: Corrected as suggested

Line 144 - remove "experimental". These adjuvants are now used in licensed vaccines and have passed the preclinical and clinical development stage.

Response: "Experimental" deleted

Line 562 - correct to "anecdotes"

Response: Corrected as suggested